# Measuring Visual Generalization in Continuous Control from Pixels

## Abstract

Self-supervised learning and data augmentation have significantly reduced the performance gap between state and image-based reinforcement learning agents in continuous control tasks. However, it is still unclear whether current techniques can face the variety of visual conditions required by real-world environments. We propose a challenging benchmark that tests agents' visual generalization by adding graphical variety to existing continuous control domains. Our empirical analysis shows that current methods struggle to generalize across a diverse set of visual changes, and we examine the specific factors of variation that make these tasks difficult. We find that data augmentation techniques outperform self-supervised learning approaches, and that more significant image transformations provide better visual generalization.

## 1 Introduction

Reinforcement Learning has successfully learned to control complex physical systems when presented with real-time sensor data (Gu et al., 2017) (Kalashnikov et al., 2018). However, much of the field's core algorithmic work happens in simulation (Lillicrap et al., 2015) (Haarnoja et al., 2018a), where all of the environmental conditions are known. In the real world, gaining access to precise sensory state information can be expensive or impossible. Camera-based observations are a practical solution, but create a representation learning problem in which important control information needs to be recovered from images of the environment.

Significant progress has been made towards a solution to this challenge using auxiliary loss functions (Yarats et al., 2019) (Zhang et al., 2020) (Srinivas et al., 2020) and data augmentation (Laskin et al., 2020) (Kostrikov et al., 2020). These strategies often match or exceed the performance of state-based approaches in simulated benchmarks. However, current continuous control environments include very little visual diversity. If existing techniques are going to be successful in the real world, they will need to operate in a variety of visual conditions. Small differences in lighting, camera position, or the surrounding environment can dramatically alter the raw pixel values presented to an agent, without affecting the underlying state. Ideally, agents would learn representations that are invariant to task-irrelevant visual changes.

In this paper, we investigate the extent to which current methods meet these requirements. We propose a challenging benchmark to measure agents' ability to generalize across a diverse set of visual conditions, including changes in camera position, lighting, color, and scenery, by extending the graphical variety of existing continuous control domains. Our benchmark provides a platform for examining the visual generalization challenge that image-based control systems may face in the real world while preserving the advantages of simulation-based training. We evaluate several recent approaches and find that while they can adapt to subtle changes in camera position and lighting, they struggle to generalize across the full range of visual conditions and are particularly distracted by changes in texture and scenery. A comparison across multiple control domains shows that data augmentation significantly outperforms other approaches, and that visual generalization benefits from more complex, color-altering image transformations.

## 2 BACKGROUND

**Reinforcement Learning:** We deal with the Partially Observed Markov Decision Process (POMDP) setting defined by a tuple $(\mathbb{S}, \mathbb{O}, \phi, \mathbb{A}, R, T, \gamma)$. $\mathbb{S}$ is the set of states, which are usually low-dimensional representations of all the environment information needed to choose an appropriate action. In contrast, $\mathbb{O}$ is typically a higher-dimensional observation space that is visible to the agent. $R$ is the reward function $\mathbb{S}$ x $\mathbb{A} \to \mathbb{R}$, $T$ is the function representing transition probabilities $\mathbb{S}$ x $\mathbb{S} \to [0, 1]$. $\phi$ is the emittion function $\mathbb{S} \to \mathbb{O}$ that determines how the observations in $\mathbb{O}$ are generated by the true states in $\mathbb{S}$. $\mathbb{A}$ is the set of available actions that can be taken at each state; in the continuous control setting we focus on in this paper, $\mathbb{A}$ is a bounded subset of $\mathbb{R}^n$, where $n$ is the dimension of the action space. An agent is defined by a stochastic policy $\pi$ that maps observations to a distribution over actions in $\mathbb{A}$. The goal of Reinforcement Learning (RL) is to find a policy that maximizes the discounted sum of rewards over trajectories of experience $\tau$, collected from a POMDP $\mathcal{M}$, denoted $\eta_{\mathcal{M}}(\pi)$: $\eta_{\mathcal{M}}(\pi) = \mathop{\mathbb{E}}_{\tau \sim \pi} [\sum_{t=0}^{t=\infty} \gamma^t R(s_t, a_t)]$, where $\gamma \in [0, 1)$ is a discount factor that determines the agent's emphasis on long-term rewards.

**Continuous Control:** Many of the challenging benchmarks and applications of RL involve continuous action spaces. These include classic problems like Cartpole, Mountain Car and Inverted Pendulum (Moore, 1990) (Barto et al., 1990). However, recent work in Deep RL has focused on a collection of 3D locomotion tasks in which a robot is rewarded for a specific type of movement in a physics simulator. At each timestep, the state is a vector representing key information about the robot's current position and motion (e.g. the location, rotation or velocity of joints and limbs). The action space is a continuous subset of $\mathbb{R}^n$, where each element controls some aspect of the robot's movement (e.g. the torque of a joint motor). There have been several standardized implementations of these environments (Schulman et al., 2015) (Brockman et al., 2016). This paper will use modified versions of those provided in the DeepMind Control Suite (DMC) (Tassa et al., 2018), built using the MuJoCo physics simulator (Todorov et al., 2012).

**Soft Actor Critic:** Soft Actor Critic (SAC) (Haarnoja et al., 2018a) is an off-policy actor-critic method that achieves impressive performance on many continuous control tasks. For each training step, a SAC agent collects experience from the environment by sampling from a high-entropy policy ($a \sim \pi_\theta(o)$) paramaterized by a neural network with weights $\theta$. It then adds experience to a large buffer $\mathcal{D}$ in the form of a tuple $(o, a, r, o')$[1]. SAC samples a batch of previous experience from the buffer, and updates the actor to encourage higher-value actions, as determined by the critic network ($Q_\phi(o, a)$):

$$\mathcal{L}_{actor} = - \mathop{\mathbb{E}}_{o \sim \mathcal{D}} \left[ \min_{i=1,2} Q_{\phi,i}(o, \tilde{a}) - \alpha \log \pi_\theta(\tilde{a}|o)) \right], \tilde{a} \sim \pi_\theta(o) \tag{1}$$

The critic networks are updated by minimizing the mean squared error of their predictions relative to a bootstrapped estimate of the action-value function:

$$\mathcal{L}_{critic} = \mathop{\mathbb{E}}_{(o,a,r,o') \sim \mathcal{D}} \left[ \left( Q_\phi(o, a) - (r + \gamma(\min_{i=1,2} Q_{\phi',i}(o', \tilde{a}') - \alpha \log \pi_\theta(\tilde{a}'|o'))) \right)^2 \right], \tilde{a}' \sim \pi_\theta(o') \tag{2}$$

The $\log \pi_\theta$ terms are a maximum-entropy component that encourages exploration subject to a parameter $\alpha$, which can either be fixed or learned by gradient descent to approach a target entropy level (Haarnoja et al., 2018b). The $min$ operation makes use of two critic networks that are trained separately, and helps to reduce overestimation bias (Fujimoto et al., 2018). $\phi'$ refers to target networks, which are moving averages of the critic networks' weights and help to stabilize learning - a common trick in Deep RL (Lillicrap et al., 2015) (Mnih et al., 2015).

**Generalization in Reinforcement Learning:** Consider a set of POMDPs $\mathcal{M} = \{(\mathbb{S}_0, \phi_0, \mathbb{O}_0, \mathbb{A}_0, R_0, T_0, \gamma_0), ..., (\mathbb{S}_n, \phi_n, \mathbb{O}_n, \mathbb{A}_n, R_n, T_n, \gamma_n)\}$. Given access to one or several training tasks $\mathcal{M}_{train} \sim \mathcal{M}$, we would like to learn a policy that can generalize to the entire

---

[1]We leave out the terminal boolean $d$ common to most implementations both for notational simplicity and because many of the DMC tasks reset after a fixed amount of time.

set. This is analogous to the way we train supervised models to generalize across an entire distribution despite only having labels for a finite number of inputs. In supervised learning, we would partition our labeled data into training and testing sets, and measure the difference in our model's performance between them. We can create $\mathcal{M}_{train}$ and $\mathcal{M}_{test}$ in a similar way (Zhang et al., 2018a). This paper focuses on the zero-shot generalization setting ($|\mathcal{M}_{train}| = 1$), and will normalize by the training return in an effort to compare the generalization of policies with different baseline levels of performance. A policy's generalization error, $E_G(\pi)$, is defined as: $E_G(\pi) \equiv (\eta_{\mathcal{M}_{train}}(\pi) - \frac{1}{|\mathcal{M}_{test}|} \sum_{i=1}^{|\mathcal{M}_{test}|} \eta_{\mathcal{M}_{test},i}(\pi))/(\eta_{\mathcal{M}_{train}}(\pi))$.

# 3 A Benchmark for Measuring Visual Generalization in Continuous Control from Pixels

## 3.1 Continuous Control From Pixels

While SAC performs well on many of the DMC tasks, it has access to state information that would be difficult to replicate in the context of real-world robotics ($\mathbb{O} = \mathbb{S}$). Ideally, we could learn a policy from imperfect or high-dimensional observations. This had led to a more challenging benchmark where the same set of tasks are learned from image renderings of the environment. Many successful approaches build around the addition of a convolutional encoder to the SAC agent architecture. The encoder attempts to process a stack of consecutive frames into a low-dimensional representation that is then treated as a surrogate state vector for the actor and critic(s). However, the RL setting's sparse gradient signal makes it difficult to train a high-capacity encoder with a limited number of samples, and baseline implementations typically fail to make progress (Tassa et al., 2018). Several recent works have addressed this problem by adding additional objectives to the encoder's gradient update. SAC+AE adds an image reconstruction term that trains a decoder network to recover the original image from the encoder's compressed representation, as in a standard autoencoder (Yarats et al., 2019). CURL uses a contrastive loss (Srinivas et al., 2020) that encourages similar representations across random crops of its input images. DBC uses a bisimulation metric to discard observation information that isn't relevant to control (Zhang et al., 2020). All three result in performance at or near the levels of state-based SAC and experiments show that the representation learned by the encoder recovers key state information.

**Data Augmentation in Deep RL:** Another recent line of work applies standard data augmentations from image processing to the training loops of RL algorithms. This simple technique can outperform more complicated approaches like SAC+AE and CURL. There are some subtleties in both the types of data augmentations that are used and how they are applied within the training update. RAD (Laskin et al., 2020) renders the environment at a higher resolution and randomly crops each observation as it sampled from the replay buffer. Data Regularized Q-Learning (DrQ) (Kostrikov et al., 2020) is a similar approach that includes the option to augment the $o'$ images separately within each timestep in hopes of computing a lower-variance target for the critic updates. Instead of upscaling, DrQ pads each image by replicating border pixels before randomly cropping back to the original size. Data Regularized Actor-Critic (DrAC) (Raileanu et al., 2020) more explicitly regularizes the outputs of the actor and critic networks to be invariant under each augmentation. It also attempts to learn which transformations to apply automatically; experiments in all of these works show this to be a surprisingly high-stakes problem, as many augmentation choices have little to no effect on learning (cf. Laskin et al. (2020) Figure 2a, Kostrikov et al. (2020) Appendix E Figure 6, and Appendix A.3.3). The search finds random crop to be the most helpful in many of the Procgen benchmark games (Cobbe et al., 2019a) and shows that it is a strong default even when outperformed by other methods. Most of the successful augmentations aside from Random Crop are color-based transformations. These include Network Randomization (NR) (Lee et al., 2019) - which uses randomly initialized convolutional layers to create visually distinct filters of the same image - and Color Jitter (CJ), which shuffles the color palette.

Instead of benchmarking NR, DrQ and RAD separately, we implement a custom version of Pixel SAC that lets us combine aspects of all three algorithms. The overall structure is similar to the originals: we use a large convolutional encoder that is only updated by the critic's gradients and copy many of the baselines' hyperparameters for the sake of comparison. Data

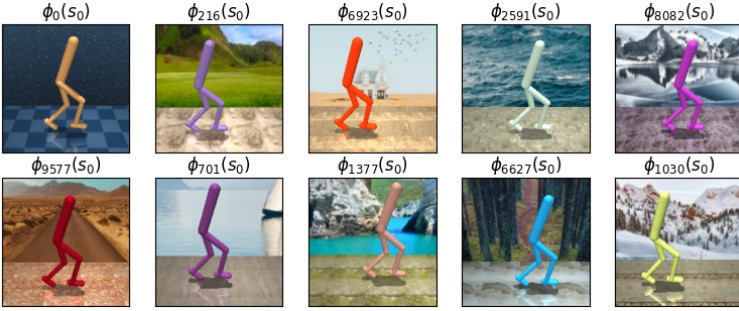

Figure 1: Example "Walker, Walk" visual seeds. The random aesthetic changes generated by DMCR allow for significant visual diversity. $\phi_0$ uses the default DMC assets.

augmentation is applied in a way that is most similar to RAD (Laskin et al., 2020). We introduce a new hyperparameter $\beta \in [0, 1]$ which controls how we mix the augmented and original batches for the actor and critic updates. $\beta = 1$ uses augmented data only; $\beta = 0$ uses the original. We set $\beta = .9$. This setup looks to alleviate the need to perform test time augmentation, even when the augmentation significantly alters the image, as in (Lee et al., 2019) (see more analysis in Appendix A.3.1). In addition, we encourage the output of the encoder, $e_\xi$ to be invariant to our augmentation pipeline $z$, by adding an explicit regularization term to the encoder loss:

$$\mathcal{L}_{encoder} = \mathcal{L}_{critic} + \lambda \mathop{\mathbb{E}}_{o \sim \mathcal{D}} \left[ ||e_\xi(o) - e_\xi(z(o))||^2 \right] \tag{3}$$

We set $\lambda = $ 1e-5, and only let gradients flow through the $e_\theta(z(o))$ pass, for stability. Appendix A.3.2 discusses this idea in more detail.

After a comparison (Appendix A.3.3), we settle on the pad/crop transformation from DrQ (Kostrikov et al., 2020) as our primary augmentation. We will refer to this algorithm as SAC+AUG. Any other transformations that are used will be added to the acronym in results and figures. For example, SAC+AUG that applies Color Jitter before the DrQ crop will be denoted SAC+CJ+AUG.

## 3.2 Observational Overfitting and Visual Generalization

While algorithms like CURL, SAC+AE and DrQ can learn successful policies from pixels, we are interested in measuring the extent to which they overfit to the visual conditions of their training environments. Observational overfitting occurs when an agent becomes dependent on features that are spuriously correlated with high reward (Song et al., 2019). Visual generalization is succinctly described as low $E_G$ across a set of POMDPs that varies primarily (or even exclusively) in $\phi$. We aim to evaluate this by creating a set of environments that vary only in their appearance, meaning that any generalization error is the result of overfitting to visual features that are not relevant to the task.

## 3.3 Expanding the Visual Diversity of DMC Tasks

Towards this goal, we introduce a modified version of the DeepMind Control Suite with an emphasis on visual diversity. Environments are given a new "visual seed", which significantly modifies the graphical appearance of the MuJoCo renderings while leaving the transition dynamics untouched. This allows the same sequence of states to be rendered in millions of distinct ways (see Figure 9). When initialized, our environments make a sequence of pseudo-random graphical choices, deterministically seeded by the visual seed, $k$, resulting in a unique appearance $\phi_k$:

1. **Floor**. The floor's pattern is sampled from a game design pack that includes hundreds of rock, grass, soil, sand, ice and concrete textures.
2. **Background**. The background is sampled from a collection of natural landscape photos.
3. **Body Color**. The color of the robot's body is sampled from the full RGB space. If there is another MuJoCo body in the scene (e.g., the glowing target location in "Reacher, Reach"), it is given a second random color.
4. **Camera and Lighting**. The camera's position and rotation are sampled uniformly within a relatively small range that varies across each domain. Lighting position is chosen similarly, along with other properties like ambience and diffusion. The limits of these distributions were determined by generating a large number of seeds across each domain and ensuring that there were sufficient variations without hiding important task information.

These simple alterations are enough to create an enormous amount of variety. Figure 1 shows a sample of 10 seeds in the "Walker, Walk" task, and Appendix A.5 contains many more examples. We also allow each type of adjustment to be toggled on or off independently, and deterministic seeding makes it easy to train or evaluate multiple agents in the same environments. Importantly, we decouple the visual seed from the random seed that is already present in DMC tasks, determining the robot's initial position. During training, each reset generates a new starting position, as usual. We re-brand the original random seed as the "dynamics seed" for clarity.

The initial release includes 10 of the DMC domains for a total of 24 tasks. A full list of the randomized elements that are relevant to each domain is provided in Table 5. We will refer to this version of DMC with expanded graphics as "DeepMind Control Remastered" (DMCR).

## 4 Experimental Results and Analysis

We evaluate several recent works in pixel-based continuous control on DMC Remastered. The experimental results are organized as follows: in Section 4.1, we test the baselines' performance on the more complicated graphics assets of DMCR under the classic setup of a single training and testing seed. In Section 4.2, we evaluate each method's ability to generalize to the entire distribution of levels. In Section 4.3, we test each factor of variation in isolation to get an understanding of the specific visual characteristics current methods struggle with the most. Then in Section 4.4, we highlight the advantages of data augmentation methods and investigate the performance of more powerful image transformations like Color Jitter and Network Randomization.

### 4.1 Training Pixel-based Agents on Complex Graphics Textures

We first look at whether agents can be successfully trained from scratch inside environments with DMCR's more distracting visuals, including changes in lighting and camera position. We train CURL, SAC+AE, and SAC+AUG on the original DMC visuals ($\phi_0$) and three representative samples from the DMCR version of "Walker, Walk" - one of the more difficult tasks in the benchmark. The CURL and SAC+AE runs use their authors' original implementations, while SAC+AUG is a mix of data augmentation methods, as discussed in Section 3.1. Results are shown in Figure 2. SAC+AE and CURL perform well across each visual seed, but SAC+AUG collapses early in training in one environment. Following this, we collect data using the original DrQ implementation and notice a similar effect. We investigate this issue further by comparing the performance of SAC+AUG and CURL in the original environment against more than 60 visual seeds from the DMCR version of "Ball in Cup, Catch." Results (shown in the bottom right of Figure 2) suggest that DMCR environment visuals are slightly more difficult than the high contrast default appearance of DMC.

### 4.2 Generalizing to the Full $\phi$ Distribution

Next, we measure the ability of agents trained in one environment to adapt to the full visual diversity of the DMCR benchmark. We train CURL, SAC+AE, and SAC+AUG agents in six tasks, using the original ($\phi_0$) visuals. The pixel-based MuJoCo literature makes an important distinction between 'training steps' and 'environment steps.' A training step is defined by the agent taking an action in the environment and storing the resulting experience in its replay buffer. The simulator repeats the action several times before returning the next observation; this 'frame skip' or 'action repeat' hyperparameter creates a discrepancy between the training step count and the total number of environment timesteps. Our experiments use the settings listed in Table 3 (see Appendix). After training, the agent is evaluated over 100 dynamics seeds in its original training environment and tested in 3 dynamics seeds across 100 different visual seeds from DMCR. Table 1 reports the mean return over each set, as well as the normalized generalization error. The results show that these methods do not learn features capable of generalizing across significant visual changes.

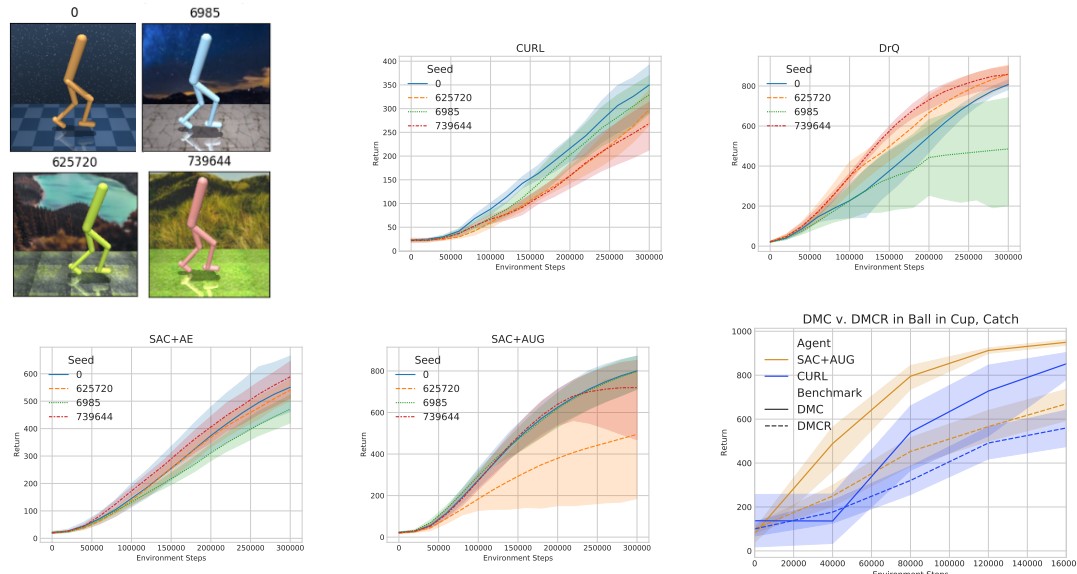

Figure 2: Pixel SAC variants trained on 4 different visual seeds from the DMCR version of "Walker, Walk". Final performance is relatively consistent, though the pure data augmentation methods are prone to occasional collapses early in training that are difficult to recover from. We also provide a comparison of the DMC and DMCR versions of "Ball in Cup, Catch" (in bottom right).

| Method: | SAC+AE | | | CURL | | | SAC+AUG | | |
|---|---|---|---|---|---|---|---|---|---|
| Metrics: | Train | Test | $E_G$ | Train | Test | $E_G$ | Train | Test | $E_G$ |
| Walker, Walk | 643.6 | 25.0 | 96.1% | 622.7 | 25.3 | 95.9% | 919.3 | 28.0 | 97.0% |
| Cheetah, Run | 387.9 | 4.2 | 98.9% | 202.3 | 2.4 | 98.8% | 712.8 | 19.8 | 97.2% |
| Hopper, Stand | 631.8 | 1.6 | 99.8% | 425.5 | 1.9 | 99.5% | 878.1 | 3.3 | 99.6% |
| Finger, Spin | 622.1 | 0.1 | 100.0% | 731.3 | 1.6 | 99.8% | 916.7 | 7.4 | 99.2% |
| Ball in Cup, Catch | 647.0 | 99.2 | 84.6% | 815.6 | 106.3 | 87.0% | 934.5 | 103.0 | 89.0% |
| Cartpole, Balance | 942.8 | 212.3 | 77.5% | 885.7 | 233.3 | 73.7% | 987.4 | 218.8 | 77.8% |

Table 1: Training and Testing returns across 6 tasks. Training scores are averaged over 100 random initializations with no visual changes. Testing scores use 3 random initializations on 100 visual seeds. $E_G$ indicates the agent's decline in performance relative to it's training environment. Results are averaged over five runs.

## 4.3 IDENTIFYING THE MOST DIFFICULT FACTORS OF VARIATION

While generalizing to the entire distribution of visuals is a desirable goal, the high level of difficulty can make it hard to differentiate between algorithms and measure gradual progress. It would be helpful to know the specific factors of variation that are most challenging. We investigate this by allowing each category of visual changes to be toggled on or off independently. Each agent is re-evaluated over 100 visual seeds, each with 3 dynamics seeds, across 7 variations of the core environment. Results for all 6 tasks are shown in Table 2. In addition to the CURL, SAC+AE and SAC+AUG agents from Section 4.2, we test the impact of Color Jitter and Network Randomization augmentations. SAC+AUG has the highest returns across every task, adding more evidence to recent findings (Laskin et al., 2020) (Kalashnikov et al., 2018) that simple augmentation techniques can outperform the more complicated auxiliary losses of SAC+AE and CURL. We find that all methods adapt well to changes in lighting conditions. Aside from SAC+AE, many agents can also handle the camera shifts and rotations. This may be because SAC+AE is the only method that does not involve a random crop, which can have a similar effect to subtle camera translations. SAC+NR+AUG and SAC+CJ+AUG are capable of ignoring changes in floor texture, and identifying the agent as it shifts color. The large improvement in generalization between SAC+AUG and and it's heavier-augmentation variants is investigated further in Section 4.4.

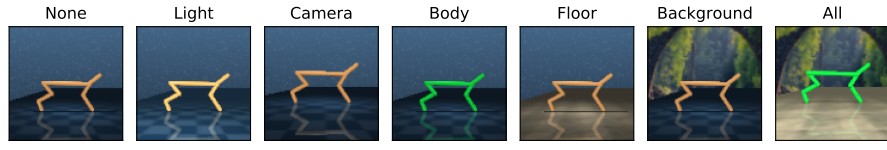

| | | None | Light | Camera | Body Color | Floor | Background | All |
|---|---|---|---|---|---|---|---|---|
| *Ball in Cup, Catch* | SAC+AE | 647.0 | 610.5 | 489.9 | 334.5 | 105.3 | 107.5 | 99.2 |
| | CURL | 822.5 | 707.0 | 766.9 | 342.3 | 169.4 | 106.5 | 109.2 |
| | SAC+AUG | 934.5 | 918.9 | 873.4 | 261.0 | 147.1 | 118.6 | 103.0 |
| | SAC+CJ+AUG | **955.6** | **954.9** | **889.6** | 928.2 | **949.3** | 509.4 | 322.1 |
| | SAC+NR+AUG | 948.5 | 947.8 | 864.9 | **937.6** | 932.6 | **582.3** | **411.2** |
| *Walker, Walk* | SAC+AE | 643.6 | 538.0 | 145.9 | 316.9 | 31.7 | 53.8 | 25.0 |
| | CURL | 622.7 | 472.8 | **532.6** | 231.7 | 25.8 | 46.3 | 25.3 |
| | SAC+AUG | 919.3 | 830.1 | 518.3 | 550.7 | 31.1 | 106.2 | 28.0 |
| | SAC+CJ+AUG | **925.1** | **905.4** | 459.6 | **852.6** | **535.3** | **240.3** | **71.9** |
| *Finger, Spin* | SAC+AE | 622.1 | 615.7 | 76.4 | 347.4 | 8.1 | 38.8 | 0.1 |
| | CURL | 731.3 | 716.9 | **521.1** | 433.7 | 29.6 | 53.6 | 1.6 |
| | SAC+AUG | **916.7** | **873.2** | 441.7 | 525.3 | 470.3 | 98.1 | 7.4 |
| | SAC+CJ+AUG | 798.5 | 796.4 | 388.8 | 760.6 | **795.7** | 274.6 | 82.8 |
| | SAC+NR+AUG | 797.6 | 788.0 | 420.6 | **779.5** | 795.5 | **332.4** | **154.7** |
| *Cheetah, Run* | SAC+AE | 387.9 | 252.9 | 36.4 | 252.6 | 65.8 | 75.9 | 4.2 |
| | CURL | 202.3 | 142.6 | 157.9 | 109.3 | 18.5 | 32.1 | 2.4 |
| | SAC+AUG | **712.7** | 321.8 | 259.3 | 471.3 | 127.9 | 310.6 | 19.8 |
| | SAC+CJ+AUG | 669.2 | **406.3** | **278.9** | **620.6** | **144.3** | **359.7** | **47.4** |
| *Cartpole, Balance* | SAC+AE | 942.8 | 810.8 | 377.2 | 492.6 | 313.8 | 254.2 | 212.3 |
| | CURL | 885.7 | 789.9 | 831.7 | 521.2 | 403.9 | 261.4 | 233.3 |
| | SAC+AUG | **987.4** | **963.0** | **878.1** | 655.2 | 565.1 | 422.2 | 218.8 |
| | SAC+CJ+AUG | 984.2 | 920.9 | 876.5 | **865.5** | **978.3** | 457.1 | 358.6 |
| | SAC+NR+AUG | 946.0 | 859.8 | 855.6 | 860.9 | 923.2 | **462.5** | **377.4** |
| *Hopper, Stand* | SAC+AE | 631.8 | 616.4 | 7.7 | 260.4 | 6.3 | 1.4 | 1.6 |
| | CURL | 425.5 | 311.9 | **202.7.4** | 72.1 | 12.4 | 4.7 | 1.9 |
| | SAC+AUG | **878.1** | 764.9 | 195.2 | 403.4 | 46.0 | 39.5 | **3.3** |
| | SAC+CJ+AUG | 848.3 | **778.6** | 188.2 | **693.4** | **144.5** | **50.5** | 1.8 |

Table 2: Return of pixel-based SAC variants in 6 tasks from DMCR. Agents are trained on the default visuals ("None"), and evaluated in versions of the environment where one aspect of the scene ("Light", "Floor", . . . ) is randomly adjusted. Testing lasts for 300 episodes across 100 distinct visual seeds. Reported scores are the average of five experiments.

## 4.4 Analyzing the Success of Data Augmentation Methods

To understand the effect that augmentations like Color Jitter and Network Randomization have on generalization, we take a closer look at their performance relative to the default DrQ crop. While SAC+AUG generalizes poorly to changes in floor texture in "Ball in Cup, Catch" (Table 2) ($E_G = 84.2\%$), SAC+NR+AUG generalizes nearly perfectly ($E_G = 1.7\%$). We speculate that the random color changes introduced by the NR augmentation reduce the variance of the encoder's state representation w.r.t changes in floor texture and color. We can measure this by using DMCR to generate a batch of observations of the exact same state, rendered with hundreds of different floor patterns. We pass these observations through the encoder of trained SAC+AUG and SAC+NR+AUG agents, and collect the resulting state representations. An encoder with perfect visual generalization would have zero output variance, meaning it predicts the same state vector for every observation. We plot the standard deviation of the agents' representations in Figure 3. Because the encoder's indices have no meaningful order, we sort them by increasing variance. We average the results of this experiment over five trained agents. Network Randomization learns a much more consistent representation of this state than the DrQ crop augmentation alone. This process is repeated for the SAC+CJ+AUG agents in "Cartpole, Balance" with very similar results.

Finally, we compare the spatial activation maps of SAC+AUG and SAC+CJ+AUG encoders in "Ball in Cup, Catch", and find that the CJ agent is much less distracted by a change in floor texture. Results can be found in Appendix A.1 Figure 4. We argue that the zero-shot generalization benchmark provided by DMCR is a better use-case for significant

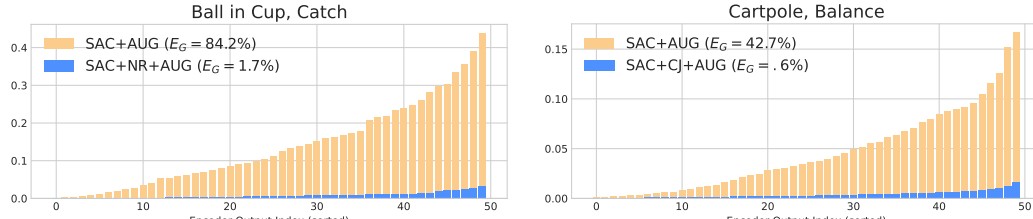

Figure 3: Standard deviation of the encoder network's output across renderings of the same state with different floor textures and colors. Indices sorted by increasing variance. Results averaged over five agents.

image alterations like CJ and NR than the typical single-environment setting, where simple random crops have become the default (Laskin et al., 2020) (Kostrikov et al., 2020). By randomizing color during training, they act as a kind of self-generated domain randomization (Tobin et al., 2017) (Mehta et al., 2019) (OpenAI et al., 2019) and regularize learning by increasing generalization across tasks at the expense of performance on the training task. This is also supported by the success of NR and CJ on Procgen (Raileanu et al., 2020) (Cobbe et al., 2019a), where success involves a similar amount of visual generalization.

## 5 Related Work

The generalization of Deep RL methods is an active area of research. Agents have been shown to overfit to deterministic environment elements (Zhang et al., 2018a), despite efforts to add stochasticity during evaluation (Zhang et al., 2018c) (Machado et al., 2017). Other work focuses on generalization across environments with different transition dynamics, particularly in the continuous control setting (Packer et al., 2019) (Zhao et al., 2019) (James et al., 2019) (Plappert et al., 2018), and recommends an evaluation protocol where agents' generalization is assessed based on their performance in a held out set of tasks (Cobbe et al., 2019b). This has led to the development of benchmarks with a large or infinite (procedurally generated) number of similar environments (Nichol et al., 2018) (Juliani et al., 2019) (Justesen et al., 2018). Notable among these is Procgen (Cobbe et al., 2019a) - a set of games meant to imitate the strengths of the Atari 57 benchmark (Bellemare et al., 2013) but with procedurally generated level layouts and visuals. Unlike the DMCR environments discussed in this paper, tasks in the Procgen benchmark have different state spaces and transition dynamics, and demand a more universal form of generalization; DMCR tests perception and observational overfitting in isolation by keeping the underlying task fixed. Procgen's discrete action space also lends itself to a different family of algorithms than are often used in the continuous control setting.

Other work has investigated visual generalization by adding spurious visual features to popular RL environments (Gamrian & Goldberg, 2019) (Roy & Konidaris, 2020) (Sonar et al., 2020). This paper, and the DMCR benchmark, is most directly inspired by (Zhang et al., 2018b), as well as experiments in (Zhang et al., 2020) and (Yarats et al., 2019), where the observations of similar continuous control tasks are processed to remove all of the floor and background pixels and replace them with natural images or video. This is usually accomplished by masking out the color of the background after the observation has been rendered. DMCR takes this idea a step further, and directly changes the graphical assets of the environment. This design gives it more control over the tasks' appearance and opens up the opportunity for changes in lighting and camera position. It is also the first time this concept has been made into a reproducible benchmark for future research.

## 6 Conclusion

In this work, we propose a challenging benchmark for measuring the visual generalization of continuous control agents by expanding the graphical variety of the DeepMind Control Suite. Our experiments show that recent advances in representation learning in RL are not enough to achieve a high level of performance across a wide range of conditions, but suggest that data augmentation may be a promising path forward.

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

## A  APPENDIX

### A.1  SPATIAL ATTENTION MAP RESULTS

Using a method described in (Zagoruyko & Komodakis, 2017), we analyze the spatial attention maps of SAC+AUG and SAC+CJ+AUG encoder networks in "Ball in Cup, Catch." The activations of intermediate convolutional layers are resized and superimposed over the original observation, giving us a sense of what the agent is paying attention to. In the training environment, both agents are fixated on the path of the ball[2], as well as the closest edge of the cup. However, SAC+AUG becomes distracted by a change in floor texture, while the Color Jitter variant remains focused on the task. This provides some insight into why SAC+CJ+AUG performs so much better than the default SAC+AUG when testing over random floor changes (see Table 2).

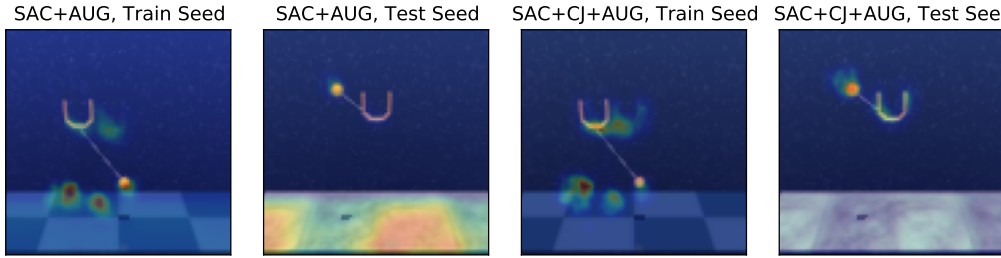

Figure 4: Spatial attention maps of trained SAC+AUG and SAC+CJ+AUG agents. Computed by taking the channel-wise average of encoder layer activations, and overlaying them on the original observation. Green and red heatmap colors indicate high levels of attention. Both agents are trained with the default checkerboard floor, but the CJ agent is less distracted by a change to a concrete texture.

### A.2  CONNECTING TO GENERAL DATA AUGMENTATION

Data augmentation synthesizes new training data from an available set of samples and provides a practical and common way to achieve better generalization. Basic data augmentation strategies like random affine and projective transformations, and color jittering, have shown significance in computer vision applications.

We can group existing data augmentation methods roughly into four groups: (1) spatial or color transformation, (2) information deletion, (3) mixing up samples, and (4) meta-augmentation.

- (1) Spatial transformation and color distortion revise some channels of original image information (Takahashi et al., 2018). This type helps the training set better simulate the real-world.

- (2) Information dropping type is more recent, including methods like cutout (Devries & Taylor, 2017) and random erasing (Devries & Taylor, 2017; Zhong et al., 2020). This kind tries to enforce the representation learning to focus on less sensitive signals, aiming for more robust models.

- (3) Mixup Zhang et al. (2018d) proposed the idea of interpolating two images and their ground truth labels to augment the training data. Mixup and its variants aim for stable model training and increased model robustness to adversarial examples.

- (4) More recently, researchers have proposed methods to automatically search for data augmentation policies with Reinforcement Learning (RL) or Evolutionary Algorithms (Zoph & Le, 2017; Lu et al., 2019; Pham et al., 2018; Xie et al., 2019).

---

[2]The high-attention zones trailing the ball's current position are a result of past timesteps in the framestack that make up this observation, which are not pictured here.

Data augmentation investigated in this paper fall into the first first types. We foresee great potentials for benchmarking the latter two groups to reinforcement learning and will investigate shortly.

### A.3 Augmented Soft Actor Critic: Details and Ablations

SAC+AUG is a custom implementation meant to combine several of the ideas from recent work on data augmentation in RL. It uses the augmentation process described in RAD (Laskin et al., 2020), which was chosen for its simplicity; observations are augmented as they exit the replay buffer, and then used in a standard SAC learning update. We make two additions: a $\beta$ parameter to control how the augmented data is mixed into the update batch, and a feature-matching loss term.

#### A.3.1 $\beta$ parameter to alleviate test-time augmentation

Several methods apply their augmentations to observations during both training and evaluation. In Lee et al. (2019), the agent's actions are an average across several different augmentations of the current observation. CURL and RAD render the environment in a higher resolution in order to efficiently perform their random crop augmentations, which requires a center crop during testing. We experiment with skipping this step by mixing un-augmented images back into the gradient update, controlled by $\beta \in [0, 1]$. However, results in Figure 5 show this to be unimportant, at least over short training runs. Our main experiments do not use test-time augmentation, and fix $\beta = .9$.

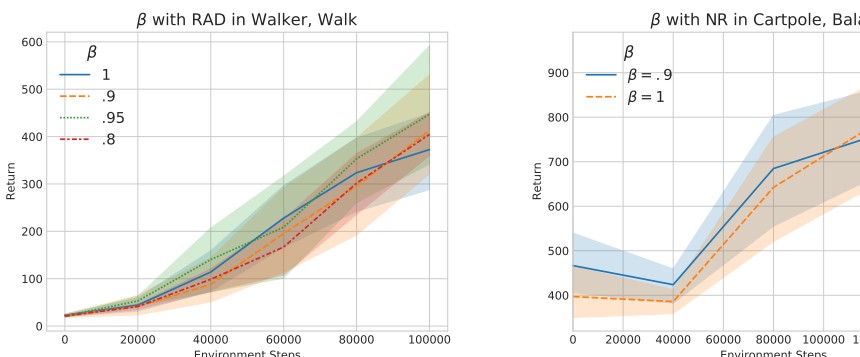

Figure 5: Testing a range of $\beta$ values on "Walker, Walk" and "Cartpole, Balance." We find there to be little difference over short training runs.

#### A.3.2 State regularization ablation

The encoder's goal is to distill the high-dimensional image observations into a low-dimensional representation that, ideally, would recover the information present in the state-based versions of these tasks. The observational changes caused by data augmentation have no effect on the true state, so we would hope that they would be equally irrelevant to the encoder's representation. Therefore, we regularize the output of the encoder $e_\xi$ to be invariant to our augmentation pipeline $z$ by adding adding a norm-based penalty that is mixed into the encoder's loss with a hyperparameter $\lambda$:

$$\mathcal{L}_{encoder} = \mathcal{L}_{critic} + \lambda \mathop{\mathbb{E}}_{o \sim \mathcal{D}} \left[ ||e_\xi(o) - e_\xi(z(o))||^2 \right] \tag{4}$$

We compare several choices of $\lambda$ in Figure 6.

#### A.3.3 Augmentation Comparison

We consider 11 augmentations to serve as the SAC+AUG default. Any randomness (e.g. where to crop, or how far to move) is kept consistent between the same batch index in the $o$ and $o'$ tensors, meaning the $o$ and $o'$ pairs that go into the critic update have identical transformations. Observations consist of a stack of 3 frames. Random elements are also con-

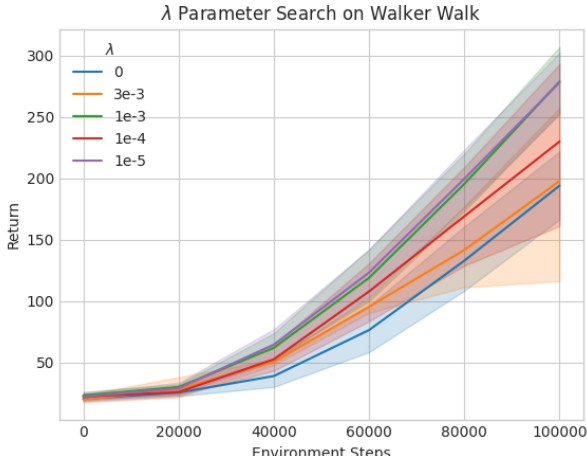

Figure 6: Testing a range of $\lambda$ values on "Walker, Walk" with $\beta = 1$ averaged over 8 trials.

sistent across frames in this stack. This requires augmentations that can be deterministically seeded.

A brief summary of each transformation:

1. **RAD** imitates the augmentation in (Laskin et al., 2020) by resizing the 84x84 observation to 100x100, and then randomly cropping back to the original size.

2. **Cutout Color** selects a rectangular patch of the image to occlude with a random color mask.

3. **DrQ** imitates the augmentation in (Kostrikov et al., 2020). The image is padded by 4 pixels by replicating the colors at the border, and then randomly cropped back to 84x84. Kostrikov et al. (2020)'s implementation uses a resampling method that turns each pixel into a non-integer float. The need for deterministic seeding prevents us from using that version, so we mimic the effect by adding small amounts of noise.

4. **DrQ No Noise** uses the same 4 pixel pad/crop as DrQ, but without the additional noise.

5. **Large DrQ** pads and crops by 12 pixels instead of 4. Figure 8 uses Large DrQ for visual effect.

6. **Large DrQ No Noise** pads and crops by 12 pixels and does not add noise.

7. **Translate** pads the image with 4 pixels of a random color, and then shifts the image within this expanded area before center-cropping.

8. **Large Translate** pads and shifts by 12 pixels instead of 4. Figure 8 uses Large Translate for visual effect.

9. **Rotate** rotates the image by one of $\{0°, 90°, 180°, 270°\}$

10. **Vertical Flip** rotates a portion of each batch by $180°$

11. **Window** selects a rectangular viewing lens to see the original image and sets all other pixels to 0.

An implementation of each transformation (many based on the RAD Procgen (Laskin et al., 2020) experiments) is available in our code release. See Figure 8 for examples.

Figure 7 shows the learning curves of all 11 augmentations over the first 100k steps of the "Walker, Walk" task, averaged over 5 trials with $\beta = .9$ and $\lambda = .0015$. In general, all pad + crop augmentations perform well; the additional noise does not appear to be an important detail. Rotations and flips slow down training significantly. Window and Cutout Color also perform poorly. This is somewhat counter intuitive, as these transformations hide information from the image much like Translate and DrQ. However, they tend to mask

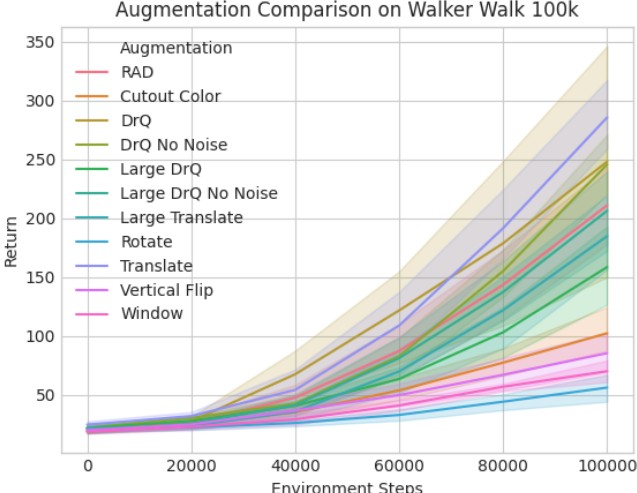

Figure 7: All the minor augmentations tested for 5 trials on "Walker, Walk" with $\beta = .9$ and $\lambda = .0015$. Figure best viewed in color.

large portions of the input, and might be making the task too difficult. This is supported by the reduced performance of DrQ and Translate when their crop factor is raised to even 12 pixels.

The limited sample size makes it difficult to distinguish between DrQ and Translate. We decide to use DrQ based on the intuition that the color-replicating padding strategy will fare better in DMCR, where it becomes more important to disregard the background.

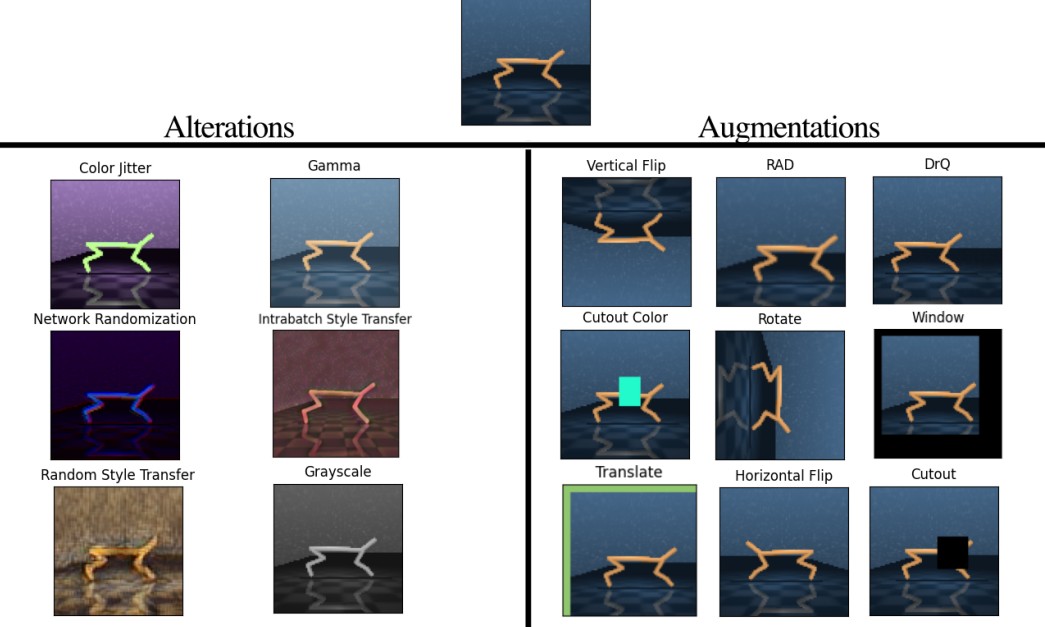

Figure 8: Examples of our augmentation implementations. We categorize the transformations into two groups. "Alterations" make significant enough visual changes that the result could conceivably be from another emission function. "Augmentations" reduce overfitting by occluding or moving the image. Our code release includes several augmentations not mentioned in the main results, which were either less effective or too computationally expensive to allow for multiple training runs.

A.4 ADDITIONAL BASELINE DETAILS AND HYPERPARAMETERS

We use the publicly available implementations of CURL, SAC+AE, and DrQ. Frame skip settings are chosen based on (Srinivas et al., 2020), and listed in Table 3 for convenience. See Table 4 for a list of other hyperparameters.

|  | Training Steps | Frame Skip (Action Repeat) |
|---|---|---|
| Walker, Walk | 250k | 2 |
| Cheetah, Run | 250k | 4 |
| Hopper, Stand | 250k | 4 |
| Finger, Spin | 100k | 2 |
| Ball in Cup, Catch | 75k | 4 |
| Cartpole, Balance | 50k | 8 |

Table 3: The frame skip settings used in all of our experiments, along with the length of training runs used to compute the results in Tables 1 and 2.

| Hyperparameter | Value |
|---|---|
| Frame Stack | 3 |
| Image Size | 84 |
| Replay Buffer Capacity | 100,000 |
| Warmup Steps | 1,000 |
| Batch Size | 256, 128 (SAC+AE) |
| Actor | 2x ReLU(FC(1024)) $\rightarrow$ Tanh(FC(out)) |
| Critics | 2x ReLU(FC(1024)) $\rightarrow$ FC(out) |
| Encoder | ReLU(Conv(32 filters, stride 2)) $\rightarrow$ 3x ReLU(Conv(32 filters, stride 1)) $\rightarrow$ FC(50) $\rightarrow$ Tanh(LayerNorm(50)) |
| Critic Learning Rate | 1e-3, 2e-4 (CURL Cheetah) |
| Actor Learning Rate | 1e-3, 2e-4 (CURL Cheetah) |
| Encoder Learning Rate | 1e-3, 2e-4 (CURL Cheetah) |
| Initial Alpha | .1 |
| Target Entropy | $-|\mathbb{A}|$ |
| Alpha Learning Rate | 1e-4 |
| Action Log Std Range | (-10, 2) |
| Encoder Tau | .05 |
| Actor Tau | .01 |
| Critic Tau | .01 |
| Gamma | .99 |
| Gradient Updates per Step | 1 |
| Update Delay | 2 |

Table 4: Hyperparameter settings used in all experiments.

A.5 DMC REMASTERED: DETAILS AND EXAMPLES

This initial version of DMC Remastered includes 10 of the DeepMind Control Suite domains for a total of 24 tasks. There are 300+ floor textures and 100+ backgrounds. Early results showed this to be more than enough of a challenge, but there is nothing to prevent (many) more images from being added in the future.

Each environment's visuals are generated by a sequence of pseudo-random decisions, deterministically seeded by the 'visual seed.' The seeds are not guaranteed to generate the same visuals across domains, although they *are* consistent across tasks within the same domain (e.g. seed 11 in Fish, Swim will look identical to seed 11 of Fish, Upright). Real-valued parameters like the camera's x, y and z coordinates or the light diffusion are sampled uniformly from a preset range of values that varies by domain. These ranges were determined by generating a large number of images and ensuring that there was sufficient variation without making the task unsolvable by hiding important information. Several of these distributions are far more conservative than they could be, in an effort to make this a useful

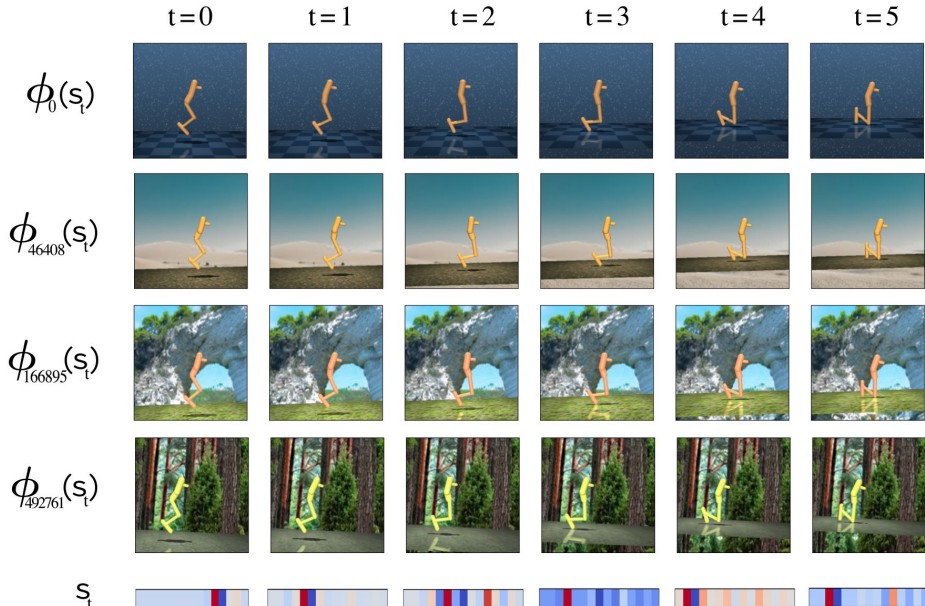

Figure 9: DMCR allows the same state trajectory to be rendered in millions of distinct visual styles. Here we show 4 examples from the "Hopper, Hop" task. Our benchmark can always recover the original DMC environment by setting the visual seed to 0.

benchmark for the abilities of current methods. In particular, the camera is restricted to minor translations and tilts, although it would be interesting to see if agents could maintain control across significantly different perspectives. Slight movements in camera position alter the raw pixel values of the observations without meaningfully changing the information within them. Moving the camera to new angles may create a more difficult challenge where the agent needs to adapt to an entirely different observation. Not every factor of variation is applicable to every domain; see Table 5 for a complete list. The 0 seed is reserved for the default DMC assets, however, some tasks alter the default camera angle in an effort to fit more of the background or floor in the frame.

The DMCR benchmark also includes a version of the environment that samples from a set of visual seeds after every reset - a convenient way to train and evaluate few-shot generalizers. Training seeds are sampled from a fixed range while testing seeds are drawn up to a large positive constant. Initial experiments suggested that successful few-shot learning requires a high-throughput training setup. We extended each of the baselines described in Sec 3.1 to collect experience from 8 actors in parallel, each interacting with a different random seed. We found that this was not enough to see positive results on all but the most straightforward domains. Due to concerns over cost, we shifted our focus to the zero-shot setting. However, the few-shot environment remains in the code release, and we hope it can be helpful in future work.

|            | Camera | Light | Body Color | Target Color | Background | Floor | Reflectance |
|------------|--------|-------|------------|--------------|------------|-------|-------------|
| Cheetah    | Yes    | Yes   | Yes        | No           | Yes        | Yes   | Yes         |
| Walker     | Yes    | Yes   | Yes        | No           | Yes        | Yes   | Limited Effect |
| Hopper     | Yes    | Yes   | Yes        | No           | Yes        | Yes   | Limited Effect |
| Cartpole   | Yes    | Yes   | Yes        | No           | Yes        | Yes   | No          |
| Finger     | Yes    | Yes   | Yes        | No           | Yes        | Yes   | No          |
| Pendulum   | Yes    | Yes   | Yes        | No           | Yes        | Yes   | No          |
| Fish       | Yes    | Yes   | Yes        | Yes          | No         | Yes   | No          |
| Humanoid   | Yes    | Yes   | Yes        | No           | No         | Yes   | No          |
| Ball in Cup| Yes    | Yes   | Yes        | No           | Yes        | Yes   | No          |
| Reacher    | Yes    | Yes   | Yes        | Yes          | No[1]      | Yes   | No          |

Table 5: **Factors of variation that are applicable to each environment.** The camera position, lighting conditions, body color and floor texture randomizations are always available. Some tasks use birds-eye-view default cameras that do not show the background. 'Reflectance' adjusts the visibility of the agent's reflection on the floor, a very small or non-existent change in most environments that is included as a way to screen for extreme overfitting. [1]Adding camera movement to the Reacher environment will show some of the background (see Figure 12).

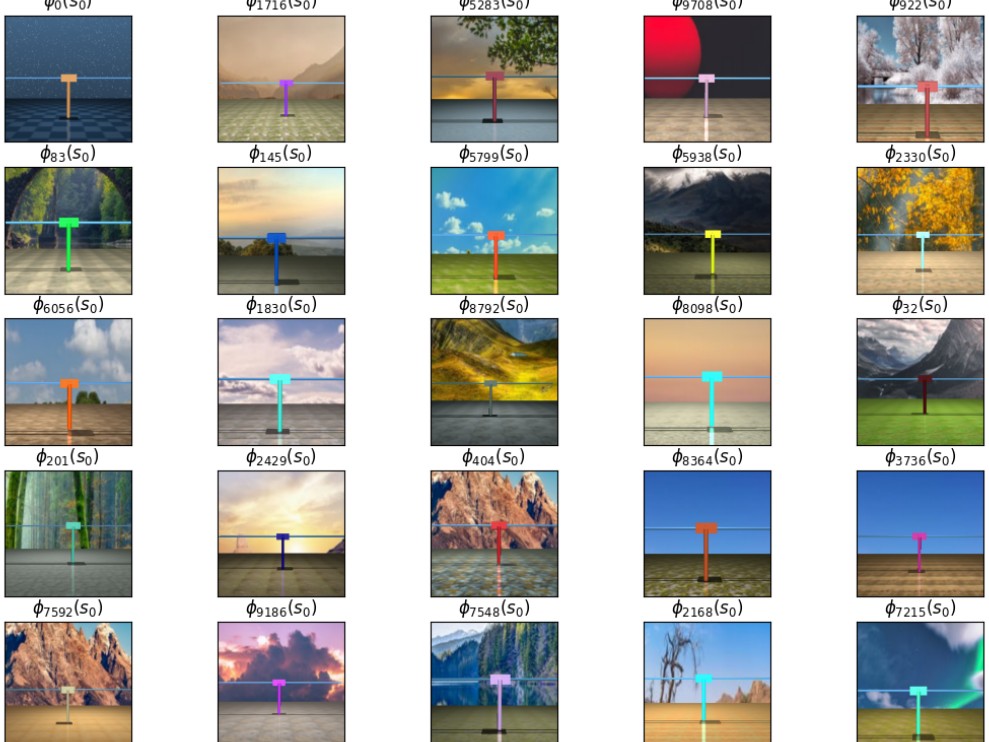

Figure 10: Example seeds from the DMCR version of "Cartpole, Swingup."

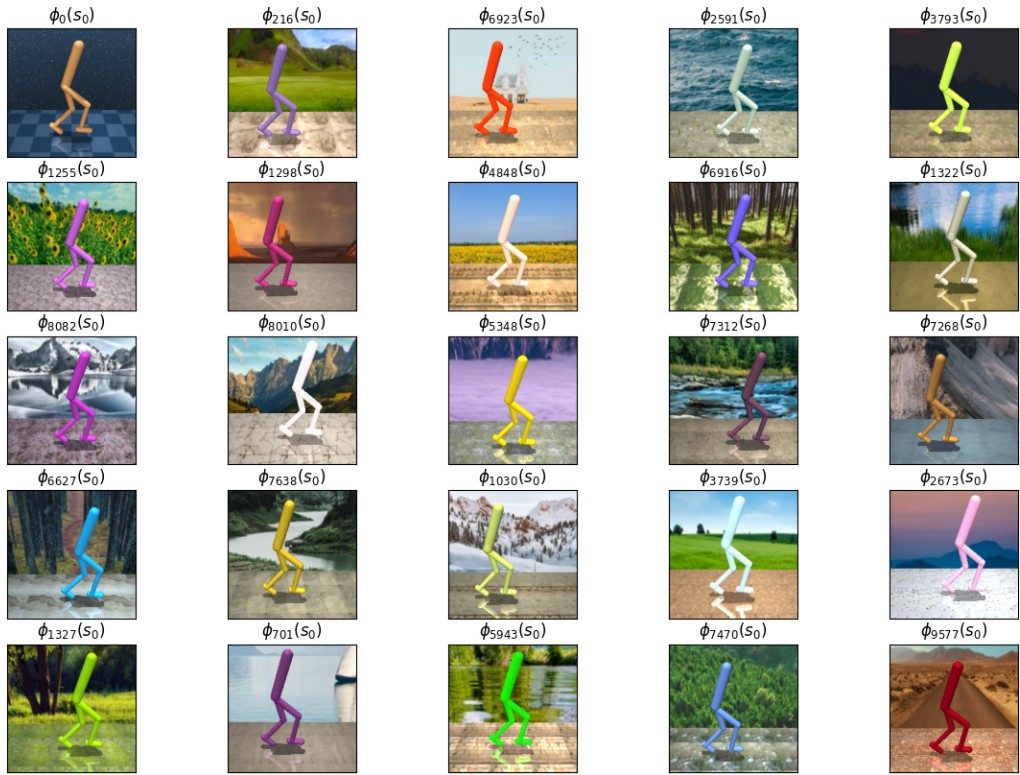

Figure 11: Example seeds from the DMCR version of "Walker, Walk."

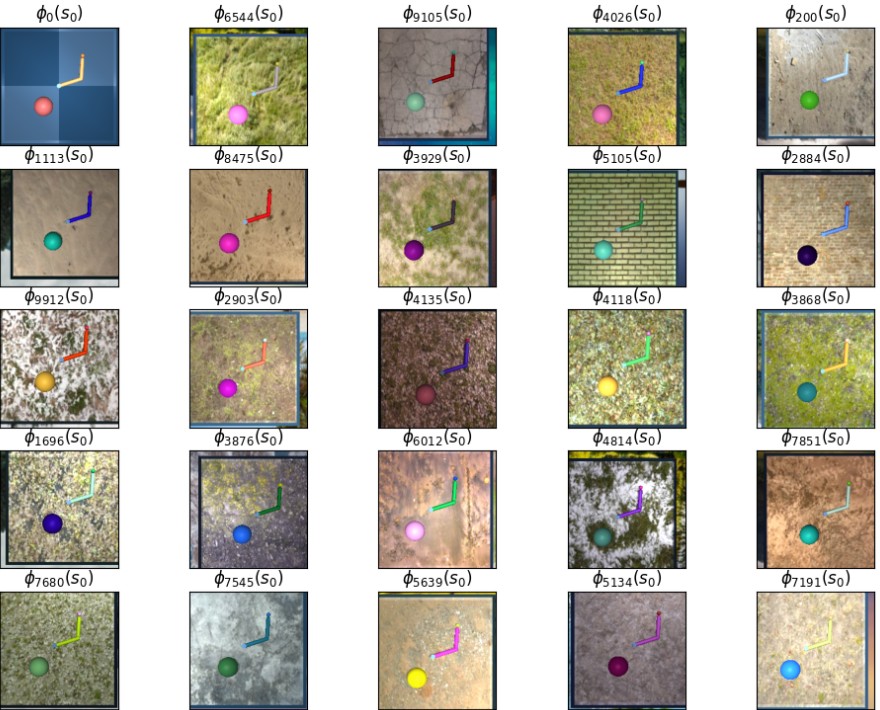

Figure 12: Example seeds from the DMCR version of "Reacher, Easy." The birds-eye view reduces visual complexity considerably. Note that the target position is given its own randomized color.

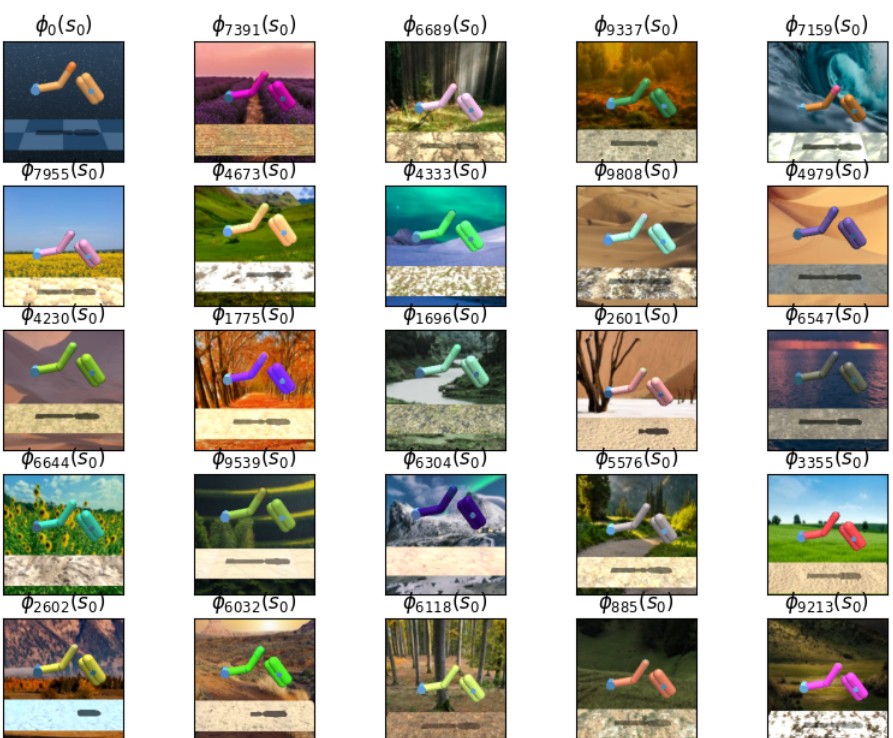

Figure 13: Example seeds from the DMCR version of "Finger, Spin."

