# OpenReview forum: "Measuring Visual Generalization in Continuous Control from Pixels"
_ICLR.cc/2021/Conference — Reject_

### Official Review · AnonReviewer3 · 2020-10-13
**Solid and useful benchmark. Some questions about clarity and novelty.**

**Rating:** 6
**Confidence:** 4

**Review:**

### **Summary and Contributions of Paper**
This paper proposes a new RL Generalization benchmark based on the DM Control Suite, where there are multiple changing backgrounds but fixed dynamics. Multiple baselines, most of which involve data augmentation for RL, are tested and shown to possess poor performance on this benchmark, which suggests that it is challenging and meaningful enough to be used for future research.

### **Strengths**
- This benchmark appears to be easy to use, and highly accessible once released.
- The baselines used are comprehensive, and contain multiple recent and competitive methods, especially in data augmentation.
- This benchmark is highly motivated, as it is important to cleanly separate the visual and dynamics factors of RL generalization, and provides good background/related work.
- Paper is written well and is easily understandable.

### **Weaknesses**
- Unfortunately, there have already been multiple benchmarks proposed with the "changing background" setting, such as in (Zhang et al 2018b, Sonar 2020). Nearly identical DeepMind Control/background benchmarks have been proposed in (Zhang et al 2020). Therefore, this paper will need to emphasize its novelty component. Some ways to provide novelty are described in the next points.
- One unfortunate trap that many RL generalization benchmark papers fall into, is the pattern of "Propose benchmark, test algorithms/hyperparameters", without providing revelations/ablations as to why such methods work conceptually, or why certain phenomena exist. Many times it is difficult to tell if problems in generalization occur specifically with the certain benchmark, or actually occur at a broad level from some deep reason. Differentiating from the "N+1-th changing background benchmark" would allow this paper to provide much more impact.
- It would be helpful if the authors could provide more insights (other than only showing reward numbers) into why overfitting occurs in this benchmark, such as e.g. 1. Saliency plots on what the RL agent focuses on, 2. Properties of the policy/SAC algorithm, such as how different data augmentations affect its optimization process.

Overall, I currently give a score of 6 (marginally above acceptance) because the usefulness/assumed accessibility of this benchmark outweighs the weaknesses, but I would be happy to increase my score if the authors address some of my concerns.

### **Clarity Questions**
- One of the most important details I was looking for was how/if the background moves compared to the agent, i.e. what is the precise definition of \phi? This is very important, as from experience in running these types of benchmarks, overfitting does not usually occur when the background moves independently to the agent (e.g. when the agent is stopped, the background can still keep moving). I assume from Fig. 8 that the background moves along with the agent, and thus forward progress (via reward function) can be spuriously correlated with the background, but could you please clarify this point?


### **References**
- https://arxiv.org/pdf/1811.06032.pdf (Zhang et al 2018b)
- https://arxiv.org/abs/2006.10742 (Zhang et al 2020)
- https://arxiv.org/pdf/2006.01096.pdf (Sonar 2020)

### **Rebuttal Update**
I thank the authors for updating their draft. I think this paper is worth accepting due to its ease of use and many different features, but I will keep my current score at a 6.

**Here's why:**

My experience in these types of benchmarks have usually shown that **completely static** backgrounds (when cleanly used - there can be very subtle things that can still correlate with progress) do not actually affect generalization, but rather, anything correlated with progress will cause overfitting. The difference between this benchmark's background and e.g. a ProcGen/CoinRun's background is that CoinRun's background still moves when the character moves (which implies a slight correlation with forward progress).

From the authors' responses to Q 3.2 and Q 3.3, I believe that the main visual overfitting is occurring with the floor tile, as it is the only spurious object that is correlated with progress. This is because, as the authors have stated, the actual background image does not actually change when the agent is moving as the background is too far away. The authors also reinforce this aspect when they added the saliency/spatial attention maps in Appendix A.1, which shows that most of the attention is focused on the floor tiles, especially on test environments.

This means that I suspect that the randomized static background portion of the benchmark does not affect generalization, and I urge the authors to rethink this portion of the benchmark.

---

> ### Author Response · Authors · 2020-11-22
> **Thank you for your reviews. Here is our Response to AnonReviewer3:**
>
> ### Q3.1: “ there have already been multiple benchmarks proposed with the "changing background" setting, such as in (Zhang et al 2018b, Sonar 2020). …. Differentiating from the "N+1-th changing background benchmark" would allow this paper to provide much more impact.”
>
> A: There are multiple benchmarks that change the background of continuous control environments. Zhang 2018b and Yarats 2019 were inspirations for our benchmark, and we try to take what makes those experiments so interesting and expand upon them. Those environments work by masking out the background of the observation after it has been rendered, usually by removing all the pixels that match the background color. Unfortunately, that approach doesn’t leave much room for further variations. Once the background is masked out and replaced, there isn’t really anything else to change. Our benchmark does not rely on separating the agent from the background after the observation has been rendered. Instead, our version actually changes the environment itself. This gives us much more control and opens up opportunities for new changes:
>
> 1. We can move the camera in relation to the agent.
> 2. We can adjust lighting intensity and angle.
> 3. The background and floor are rendered in 3D space, which adds depth and lets the replacement images shift according to the agent’s actions.
>
> These changes would be difficult or impossible to implement with the background-replacement approach, and we have just scratched the surface of what’s possible. We also adjust the agent’s body color, which is a new addition, even though it could have been done with pixel replacement.
>
> We’ll note that Zhang 2020 is recent work, and admit that we missed it until very late in our writing process (although we did add it as being a key related work). However, the environment being used by Zhang 2020 is very similar to the Yarats 2019 version, so the previous discussion still applies. The main difference is that Zhang 2020 includes a version with a natural video background. We have some concerns that motion in the video that is uncorrelated with the agent’s actions may reduce the threat of observational overfitting (as you also point out). We think the Deep Bisimulation for Control (DBC) method discussed there is a great idea and are optimistic about how it will perform in DMCR. There was some hope that we could get DBC re-implemented and benchmarked before the discussion period was over. We ultimately decided that there weren’t quite enough details in the Zhang 2020 paper to get a reproduction we were confident in, at least until the open source version is released.
>
> Finally, we hope that our version can serve as a standardized and open-sourced benchmark for future research, rather than a side experiment in an algorithm-focused paper.
>
>
> ### Q3.2: “provide more insights (other than only showing reward numbers) into why overfitting occurs in this benchmark, such as e.g. 1. Saliency plots on what the RL agent focuses on, 2. Properties of the policy/SAC algorithm, such as how different data augmentations affect its optimization process.”
>
> A: We’ve added a spatial attention map experiment (Appendix A.1, page 12), which we think adds some insight into why performance degrades when the agents leave their training environment.
>
> We have also added a second example in our investigation of how data augmentations impact generalization by reducing variance across visual seeds. As far as how augmentations affect the optimization process, we compare the impact of different data augmentations on SAC’s learning curve in Appendix A.3.3. The conclusion is that the choice of augmentation is an important detail, and small crops and translations tend to be the most effective.
>
> ### Q3.3: “how/if the background moves compared to the agent, i.e. what is the precise definition of \phi? ….  I assume from Fig. 8 that the background moves along with the agent, and thus forward progress (via reward function) can be spuriously correlated with the background, but could you please clarify this point?”
>
> A: The background changes are implemented by retexturing the night sky of the default DMC environments. The backgrounds are so far away that they don’t scroll as the agent walks or runs, although that may be a good feature for the future. The floor tile movement is based directly on the agent’s actions. Because the camera is often fixed to the agent, the background and floor will shift as the agent moves or falls over in order to keep the agent centered on the screen.

---

### Official Review · AnonReviewer1 · 2020-10-26
**A nice dataset and idea but maybe a bit too simplistic?**

**Rating:** 6
**Confidence:** 3

**Review:**

This paper presents a new dataset for continuous control tasks based on the DeemMind control suite, but with varying backgrounds, colors and lighting/camera conditions. Instead of the fixed appearance in the original suite the frames from this dataset contain a lot of varying structure. The purpose of the dataset is to measure the generalization and transfer abilities of pixel-based continuous control agents across different visual variations.
The paper presents a comparison between several different representation learning method for CC models and concludes that augmentation plays a big role in making such models generalize well.

All in all I think this is an interesting contribution and the dataset may be useful for the purpose it was created, but I have several reservations.

First, I would say that this is still a rather simplistic notion of visual variability - especially the choice for not changing the camera positions in a substantial manner - this makes the task of the visual representation much easier and by "ignoring" color (as would be the result of color augmentation) most of the generalization problem really does go away. Demonstrating the ability of methods to be robust to more severe changes in the viewpoint would be a harder and more convincing test case (as it affects, directly, some task-relevant properties which can be extracted from the scene). Also - maybe changing the actual *appearance* of the limbed model (say, by skinning it) would have been more interesting - the underlying dynamics would stay the same but the appearance would change greatly while still be a deterministic function of the underlying state.
Finally, in terms of the baselines used and the results shown - I think it would have been good to show another RL method other than SAC - just to make sure results (in terms of augmentation and generalizablity) to not depend greatly on the specific choice of underlying RL algorithm.

To conclude - I think this is a nice contribution with a potential to be even nicer.

*Post Rebuttal*

The authors have mostly addressed my concerns - I still think SAC should not only be the only RL method in the paper, and though I'm still not convinced more drastic camera variations wouldn't make this more interesting I think all in all this is a decent contribution and probably should be accepted.

---

> ### Author Response · Authors · 2020-11-22
> **Thank you for your reviews. Here is our Response to AnonReviewer1:**
>
> ### Q1.1: “ in terms of the baselines used and the results shown - I think it would have been good to show another RL method other than SAC ”
>
> A: It’s true that all of the baselines in our results use a modified version of SAC, even though the techniques are broad enough to be applied to any off-policy model-free method. Unfortunately, this is a problem we inherit from almost all of the key prior work, including RAD, DrQ, CURL, DBC, and SAC+AE. This line of work seems to have converged on SAC as its default learning method, at least when we’re dealing with continuous action spaces (CURL tests DQN on the discrete actions of Atari). Because the original goal of these papers was to close the gap between state and image-based agents, SAC is a pretty well-motivated choice; its performance generally represents the high water mark of off-policy methods. TD3 would have been a good alternative, but it’s so similar to SAC that we wouldn’t expect the results to be all that unique. We certainly agree that it would be nice to verify that these kinds of representation-learning techniques can improve a range of RL methods, but we think those comparisons may have been more useful in the original work than in our benchmark. As things are, we feel like it’s best to stay faithful to the official SAC-based versions of the baselines.
>
>
> ### Q1.2: “- especially the choice for not changing the camera positions in a substantial manner - ”
>
> A: Yes, the camera initialization settings are much more conservative than they could be. We use subtle translations and tilts that shift the agent to different parts of the frame while trying not to crop out any of its movement. The intention here was to test overfitting by seeing if performance significantly changed when the camera was in a different position, but still showed roughly the same angle. That way we’re looking at essentially the same observation but presented with different raw pixel values. An alternative would be to have the camera swing all the way around the model, but there may be many views that hide information or present it in a fundamentally different way, like not being able to see the full stride of the Walker’s legs or having the Cheetah run from right to left instead of left to right. We think the limited-movement camera option is a useful test for observational overfitting, but we will look to add an additional “factor of variation” parameter for more extreme changes in perspective.
>
>
> ### Q1.3: “- maybe changing the actual appearance of the limbed model (say, by skinning it) would have been more interesting”
>
> A: This is a great suggestion. It would be interesting to change the models in a way that does not affect their motion, like swapping them out for a higher-polygon version with different details or textures. We will add this to the long-term roadmap.

---

### Official Review · AnonReviewer4 · 2020-10-28
**This paper presents a "remastered" version of the Deep Mind Control suite that allows for testing of visual generalization. The paper presents an interesting augmentation to the DMC environment.**

**Rating:** 5
**Confidence:** 3

**Review:**

This paper presents a "remastered" version of the Deep Mind Control suite that allows for testing of visual generalization to a number of environmental changes (e.g., pattern/texture of the floor, background, and lighting).

I will begin my review by stating that I have somewhat limited knowledge of soft actor critic, Deep Mind Control and some of the most recent related work in this space and my review should be taken with that caveat.  However, the proposed forms of data/simulation augmentation that are proposed seem quite useful as a way of moving the Deep Mind Control suite from towards a more realistic and richer set of visual data. While this paper seems positioned as a more systematic baselining and data set paper and the novelty from an methodological perspective is somewhat low.  I think these are helpful in increasing how challenging the baseline is and providing more sight about how well the current methods perform.

Having said that I think there are some ways the paper could be improved.  The link to the anonymous page wasn't working for me, I tried several PDF viewers but couldn't access it.  Also the image assets did not seem to be included in the supplementary material.  The main reason I was looking for these is that the details in section 3.3. are quite limited.  If possible I would like to be able to gain more insight about the diversity of the types of augmentation.  I appreciate that spaces is limited in the paper but showing an image with small thumbnails seems like it could be an option.   I link to the assets would be very helpful.

Another element that is a little unclear to me is how the backgrounds change as the characters move.  Are these full 3D environments or 2D backgrounds?  I am referring mainly to the "Background" elements. By the description it seems that they are photo graphs and therefore they don't capture 3D elements.

The literature review around the use of simulation and/or graphics to created augmented data for interrogating models is a little bit limited.  I understand that these augmentations might be novel in the domain of RL, however, I think it is worth noting how adding conceptually this is something that has been done in past in several computer vision domains/tasks.  Not that that necessarily diminishes the utility of the proposed dataset.

Tables 1 and 2 don't have clearly labeled units.  This might seem minor but is a constant frustration for readers.
For Figure 2. The it isn't clearly indicated what the error regions represent or the units of the axes.

To summarize, I think that this work is a helpful contribution, albeit with somewhat limited innovation.  I think more information about the data augmentation and positioning in the space of other uses of graphics/simulation for testing and augmenting vision data might be helpful. I would argue the paper as it stands is borderline and needs some improvements, but those could be achieved with revisions making it a good contribution.

---

> ### Author Response · Authors · 2020-11-22
> **Thank you for your reviews. Here is our Response to AnonReviewer4:**
>
> ### Q4.1: “The link to the anonymous page wasn't working for me, I tried several PDF viewers but couldn't access it. Also the image assets did not seem to be included in the supplementary material… the details in section 3.3. are quite limited. If possible I would like to be able to gain more insight about the diversity of the types of augmentation. I appreciate that spaces is limited in the paper but showing an image with small thumbnails seems like it could be an option. I link to the assets would be very helpful.“
>
> A: Sorry about the link issue. In the final version, the link will take you to the code release. For this review, we removed the url to keep our submission anonymous and uploaded the code as part of the supplementary material. However, the image assets pushed the total file size above ICLR’s upload limit, so we had to leave them out.  The background photos are nature landscapes downloaded from free stock photo websites. The floor textures are from tiled game design assets. There are more than 300 floors and 100 backgrounds, which are totally arbitrary numbers; we realized that was more than enough variety to make the task difficult, but more could easily be added. We don’t have a great way to get you the exact images anonymously. However, the choice of image isn’t really the focus. We prioritized variety, and opted for natural photos to add detail/complexity. The content of the photos isn’t super relevant and they were mainly chosen for aesthetic purposes. If by the "diversity of the types augmentation" you mean the different kinds of data augmentation, we have some examples in Figure 8 on page 15. If you mean the different types of environment variation, Figures 10, 11, 12, and 13 at the end of the paper give a pretty good sense of the diversity DMCR can generate. Figures 10, 11, and 13 show off some of the background image options, while Figure 12 highlights the range of floor textures.
>
>
> ### Q4.2: “Another element that is a little unclear to me is how the backgrounds change as the characters move. Are these full 3D environments or 2D backgrounds?“
>
> A: The environments are technically 3D, in the sense that the agent models, floor and background positioning have a depth component. However, the background images themselves are just 2D pictures. The background changes are implemented by retexturing the night sky of the default DMC environments. The backgrounds are so far away that they don’t scroll as the agent walks or runs, although that may be a good feature for the future. The floor tile movement is based directly on the agent’s actions. Because the camera is often fixed to the agent, the background and floor will shift as the agent moves or falls over in order to keep the agent centered on the screen.
>
>
> ### Q4.3: “The literature review around the use of simulation and/or graphics to created augmented data for interrogating models is a little bit limited.”
>
> A: We have added a more general background on data augmentation in the revised draft Appendix Section A2:  Connecting to General Data Augmentation.
>
>
> ### Q4.4: “Tables 1 and 2 don't have clearly labeled units. This might seem minor but is a constant frustration for readers. For Figure 2... it isn't clearly indicated what the error regions represent or the units of the axes.”
>
> A: All of the tables and plots are showing the average return in the environment, which is the sum of all the rewards accumulated during an episode (the “total score” of the agent). Unfortunately, the scale of those rewards is only meaningful in relation to the performance of the other methods. To give some more intuition: most DMC environments use a binary +1 or +0 reward per timestep, given out based on whether the agent is meeting a hard-coded goal. For example, the “Walker, Walk” agent gets +1 for each timestep it is walking above a certain speed, and +0 otherwise. This means the maximum score is usually equal to the maximum timesteps per episode. In our paper (and in most DMC-based work) that number is 1000. So when you see SAC+AUG consistently scoring in the high 900s, that is essentially a perfect score. Not all of the environments are the same difficulty (based on results in wider literature). Cartpole is considered to be one of the easiest, while Cheetah and Walker are more difficult, and there are some tasks that are still difficult to solve even when given access to low-dimensional state information. Several of those environments are included in our DMCR benchmark but not listed in the paper’s results. The error bands in the plots indicate the 95% confidence interval over multiple runs of the learning algorithm. We list the sample sizes in the caption of each figure. These methods tend to have high variance, both because of the randomness involved in selecting actions and optimizing over batches from the replay buffer, and because of the relationship between how the policy is performing and how new data is being collected.

---

### Official Review · AnonReviewer2 · 2020-10-31
**Official Blind Review #2**

**Rating:** 6
**Confidence:** 3

**Review:**


This paper extends the DeepMind Control Suite benchmark by adding a series of visual variations across different tasks (e.g. lighting, color, background, textures, camera angles etc.). They also contrast a few recent self-supervised learning and data augmentation RL methods and measure these agents’ ability to generalize and transfer across a variety of visual variations provided by their benchmark. They also investigate different aspects of the variation in the scene that these agents seem to be most affected by.

Overall, this paper is clearly written and the analysis is very thorough and well motivated. However the results in my view are not that surprising: the observation that agents that have been trained only on a 'single' visual variation of an environment overfit to the visual features and fail to generalize outside of this distribution is not particularly novel. While I appreciate the amount of work that has been done here and the helpful analysis, contrasting different state of the art methods on self supervised learning, I have some questions/concerns:

1. There are a variety of existing benchmarks that allow for variation of the environment visual properties as well as variation in task distribution (e.g. [RLBench](https://arxiv.org/abs/1909.12271) & [OpenAI Fetch environments](https://arxiv.org/abs/1802.09464)). While Control Suite is one of the most widely used benchmarks for contrasting continuous control methods, I’m not convinced that it is on its own the best setting for studying visual generalization of agents as the majority of the tasks involve only a single agent/object and the appearance of agent/object in the environment is often not relevant to the task (making it a rather extreme example). I believe in order to better understand which methods allow for better generalization, one needs to combine tasks with different properties.
2. To my understanding the agents are trained only on a single instance of visual properties of the environment while tested on a large range of variations in the visual properties of the environment at test time. Maybe a more realistic test would be to allow a small variation of the environment during training time and then assess the generalisation gap?
3. Related to my first point, I would like to perhaps better understand what is the desired outcome for this benchmark. I am concerned that in order for some method to avoid overfitting to spurious correlations in such tasks, this might result in some trivial task-specific heuristics for augmenting the data, which would perhaps solve this benchmark while still failing to generalize to more interesting/complex settings involving many objects where some visual properties of the environment are indeed task relevant. Could you comment on that?

In summary, I enjoyed reading this paper and I believe the authors have done a nice job of contrasting different self-supervised learning algorithms in a challenging benchmark. I’m not fully convinced of the impact of the benchmark on future research but I think there are still interesting takeaways that would be valuable for the community.

---

> ### Author Response · Authors · 2020-11-22
> **Thank you for the reviews. Response to AnonReviewer2:**
>
> ### Q2.1: “Maybe a more realistic test would be to allow a small variation of the environment during training time and then assess the generalisation gap? “
>
>  A: We also agree that the few-shot setting is more realistic and maybe more interesting. This was the main focus for much of our project. We ran many experiments where the agents were given a version of the environment that picks a new visual seed after each reset, with the training environment sampling from a fixed (smaller) set of seeds and the testing environment sampling from a much larger number. Initial results were underwhelming - it didn’t seem like access to multiple visual seeds was improving generalization error. This is likely because the default data collection process of sampling one transition in one visual seed per training step is too much of a bottleneck to learn a generalizable policy. We attempted to address this by running multiple (4-8) actors in parallel, but this only showed positive transfer results in the simplest environments (mainly CartPole). Given the success of few-shot agents in benchmarks like Procgen, we think few-shot DMCR training is very achievable, but may need a higher-throughput training setup. Due to the large increase in training cost that comes with multiple actors, we decided it was best to focus on zero-shot generalization and examine it in more detail. However, the few-shot training and testing environments are still included in our benchmark, and we think they will be useful in future work. We have included a note about this in the DMCR Details section of the Appendix (A.5, last paragraph of page 16).
>
> ### Q2.2: “There are a variety of existing benchmarks that allow for variation of the environment visual properties as well as variation in task distribution (e.g. RLBench & OpenAI Fetch environments). ”
>
>  A: Yes, there are certainly plenty of other benchmarks for measuring generalization in RL, including RLBench and Fetch (now cited) as well as those that we mentioned in our Related Work section (Procgen, Obstacle Tower, Gym Retro, …). These tasks usually involve visual generalization, as there are many different visual features that can change without affecting the underlying task. However, they also make changes to the level layout or other environment parameters like the agent's movement or objective. By altering both the underlying MDP and the emission function, it becomes harder to isolate the problem that our paper is interested in, which is more about robust perception than task generalization. That is why we focus on changing the observation without changing the state space or transition dynamics. In principle, we could have picked any benchmark as our starting point. We felt that the DeepMind Control Suite’s widespread use in the development of new control algorithms made it a reasonable choice.
>
> ### Q2.3: “ I would like to perhaps better understand what is the desired outcome for this benchmark. I am concerned that in order for some method to avoid overfitting to spurious correlations in such tasks, this might result in some trivial task-specific heuristics for augmenting the data... ”
>
> A: The purpose of our benchmark is to motivate improvement in perception in reinforcement learning. An agent that could successfully transfer its knowledge between all of these visual seeds would have an understanding of the environment that would let it operate in unfamiliar or unforeseen visual conditions, which we think is an important requirement for real-world applications.
>
>  We’d like to avoid having any kind of basic augmentation solve this benchmark, to make sure it’s a helpful indicator of real progress. One concern we’ve had is that the color changing augmentations are overly effective because so much of the visual variety comes from changes in floor and background that are primarily color-based. We have already taken steps to address this by adjusting the resolution of our floor textures in some environments to better emphasize texture in addition to color. In general, we will continue to update this benchmark to be a useful challenge for current techniques.

---

### Author Response · Authors · 2020-11-22
**Summary of Changes we made in the revision**

Thank you to all the reviewers for the constructive comments! We have updated our draft with clarifications and several new experiments. We will address the specific concerns of each reviewer in individual replies, but here we list a summary of all the changes to the paper:

First, a few changes that are the result of continued experimentation after the submission deadline:

1. We wanted to get a more thorough comparison of the gap in difficulty between DMC and DMCR that was discovered in Section 4.2 (where we are training and testing in the same visual seed). That section now includes an additional experiment where we run CURL and SAC+AUG agents on the DMC (visual seed = 0) and DMCR (visual seed is randomly selected) versions of the same task with a much higher seed count (N=60) than was feasible in the original experiments. The results suggest that the more complicated graphics assets of DMCR are slightly more difficult than the high contrast DMC defaults, and the sample size helps clarify that this effect holds over multiple algorithms rather than just the augmentation methods. The final plot is now in the lower right of Figure 2 (page 6).

2. Cleaned up small evaluation/spreadsheet issues in a few of the trials that make up Tables 1 and 2. This does not meaningfully change the results. The biggest changes are an increase in SAC+AE’s performance in “Ball in Cup, Catch” (which still leaves it in last place) and a decrease in CURL’s performance in the full variation of “Walker, Walk”, which makes SAC+CJ+AUG the new top performer.

3. We repeated the encoder variance experiment (Section 4.4, page 8) with a second environment and augmentation combination, and made that figure more readable in general.

4. Grammar and readability revisions.

Next, we list changes specifically requested by the reviewers, or inspired by their comments:

5. Added a spatial attention map experiment (suggested by R3). We show that augmentation-based agents trained with Color Jitter are much less distracted by changes in floor texture. We hope this adds some insight into why some agents generalize so much better than others. The result fits well into the narrative of Section 4.4, but space constraints forced us to defer the figure itself to Appendix A.1 (top of page 12).

6. Added more discussion about the variation in camera angle to Appendix A.5 (based on comments of R1). Appendix A.5 now also includes a discussion of DMCR’s few-shot generalization features (based on comments of R2).

7. Added Appendix A.2, connecting data augmentation in RL to the wider augmentation literature (based on comments of R4).

8. Added another clarifying comment about our benchmark in relation to Zhang 2018b and Yarats 2019 and cited Sonar 2020 (based on comments of R3) as well as RLBench and Fetch (based on R2). We tried to emphasize that the differences in implementation open up new opportunities and allow us to expand on the ideas in those papers (see second paragraph of Related Work, page 8). However, we’re really up against the space limit on that final page. Please see our longer-form replies to R2 and R3 below, because we think this is an important point to discuss.

Finally, the reviewers made several suggestions about how the benchmark itself could be improved. We appreciate the feedback and creative input. Unfortunately, it isn’t feasible to implement and benchmark all the features in time for the discussion deadline. So while we don’t expect those improvements to impact our paper in time for this conference, we look forward to expanding the benchmark in the months to come.

---

### Decision · Program_Chairs · 2021-01-07
**Final Decision**

**Decision:**

Reject

**Comment:**

The paper proposes a modification to the DeepMind Control Suite to measure generalization with respect to visual variation. The authors run baseline experiments against their new benchmark and discover, unsurprisingly, that RL agents learning from visual observations overfit to spurious details of the observations.

Reviewers generally found the work to be clearly written, and the experimental analysis to be thorough and well done, though concerns about the rather simple nature of the visual augmentations persisted even after updates and author rebuttals. There were also concerns that by focusing only on Soft Actor Critic in the experiments.

3 of 4 reviewers felt the work met the acceptance bar, albeit only marginally. The dissenting reviewer's concerns centered on clarity (many specific issues appear to have been remedied), the relatively limited nature of the augmentations, and the fact that reviewers were not given access to the code.

While the submission has potential, improvements needed are not minor, and given the short process, we can only accept papers as is, rather than expecting certain changes. Please take the reviewers' comments into consideration as you revise and resubmit to a future venue.